# Editable Noise Map Inversion: Encoding Target-image into Noise For High-Fidelity Image Manipulation

**Mingyu Kang**[1]   **Yong Suk Choi**[2]

## Abstract

Text-to-image diffusion models have achieved remarkable success in generating high-quality and diverse images. Building on these advancements, diffusion models have also demonstrated exceptional performance in text-guided image editing. A key strategy for effective image editing involves inverting the source image into editable noise maps associated with the target image. However, previous inversion methods face challenges in adhering closely to the target text prompt. The limitation arises because inverted noise maps, while enabling faithful reconstruction of the source image, restrict the flexibility needed for desired edits. To overcome this issue, we propose Editable Noise Map Inversion (ENM Inversion), a novel inversion technique that searches for optimal noise maps to ensure both content preservation and editability. We analyze the properties of noise maps for enhanced editability. Based on this analysis, our method introduces an editable noise refinement that aligns with the desired edits by minimizing the difference between the reconstructed and edited noise maps. Extensive experiments demonstrate that ENM Inversion outperforms existing approaches across a wide range of image editing tasks in both preservation and edit fidelity with target prompts. Our approach can also be easily applied to video editing, enabling temporal consistency and content manipulation across frames.

## 1. Introduction

Text-to-Image diffusion models have demonstrated impressive performance in high-quality image generation (Rombach et al., 2022; Saharia et al., 2022; Ramesh et al., 2022; Podell et al., 2023). Building on this success, recent research has extended the capabilities to text-guided image (Hertz et al., 2022; Brooks et al., 2023; Tumanyan et al., 2023; Kawar et al., 2023; Cao et al., 2023) and video editing (Wu et al., 2023; Qi et al., 2023; Liu et al., 2024) by leveraging these diffusion models. An important step in text-driven image editing involves inverting a given image into a sequence of noise vectors, termed noise maps. These noise vectors are used for editing. The goal of inversion is to produce editable noise maps that preserve the content of the original image and exhibit high editability.

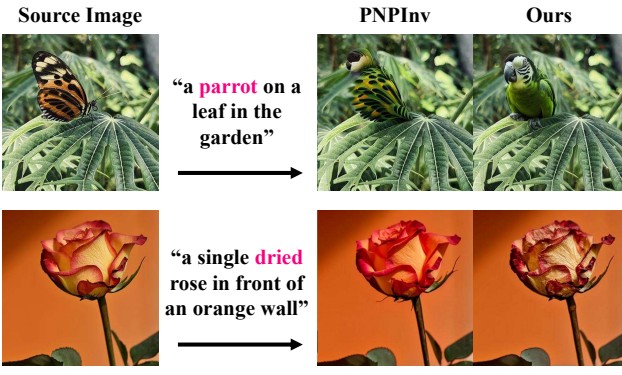

Source Image    PNPInv    Ours

"a parrot on a leaf in the garden"

"a single dried rose in front of an orange wall"

*Figure 1.* **ENM Inversion for real image editing.** PNP Inversion (PNPInv) preserves the structure and content of the original image but shows limited editing capability. Our method preserves the details of source image and enables precise modification.

In practice, many inversion methods rely on Denoising Diffusion Implicit Models (DDIM) inversion (Song et al., 2020) to obtain the latent noise representation of an image and then apply their proposed editing process. However, DDIM inversion often introduces approximation errors, resulting in noticeable inconsistencies between the reconstructed and original images. These errors are further amplified by the use of Classifier-Free Guidance (Ho & Salimans, 2022). To address this challenge, Null-Text Inversion (NTI) (Mokady

[1]Department of Artificial Intelligence, University of Hanyang, Seoul, Korea [2]Department of Computer Science, University of Hanyang, Seoul, Korea. Correspondence to: Yong Suk Choi <cys@hanyang.ac.kr>.

*Proceedings of the $42^{nd}$ International Conference on Machine Learning*, Vancouver, Canada. PMLR 267, 2025. Copyright 2025 by the author(s).

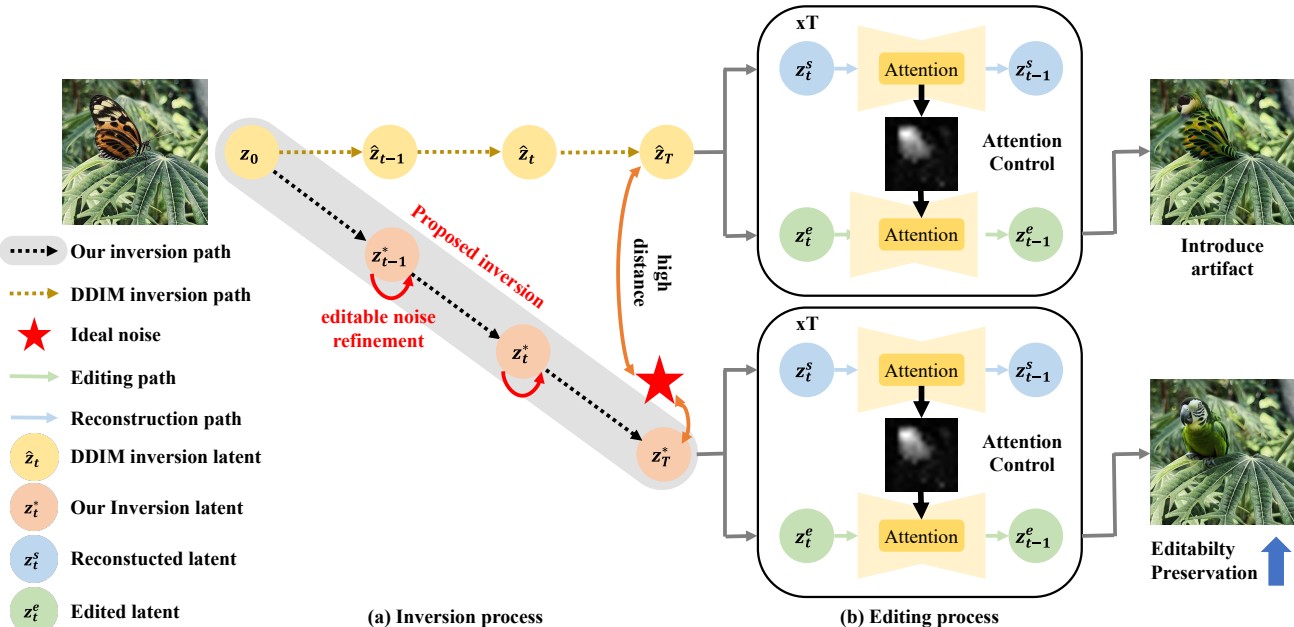

*Figure 2.* **The pipeline of image editng.** (a) DDIM Inversion transforms the input image into noise maps that allow reconstruction of the original image. Additionally, ENM inversion minimizes the gap with ideal noise by applying editable noise refinement, enabling improved reconstruction and editability. (b) Utilizing attention control, the attention maps from the reconstruction path are transferred to the editing path. Our inversion enhances both editability and preservation, resulting in the desired output image.

et al., 2023) optimizes the null-text embedding to improve reconstruction quality. Negative-Prompt Inversion (NPI) (Miyake et al., 2023) attempts to approximate the optimized null-text embedding, thereby reducing inference time. PNP Inversion (Ju et al., 2024) more efficiently preserves the essential content in the source image by adding the differences between the inverted and reconstructed noise maps. Furthermore, several works, such as fixed-point iteration methods (Pan et al., 2023; Meiri et al., 2023), reduce the approximation errors at each timestep by optimizing implicit function.

Despite these advancements in diffusion inversion, editability remains a significant challenge. The primary reason is that the noise maps obtained through DDIM Inversion are designed to reconstruct the source image rather than generate edited output. That is, the target image is not "imprinted" onto the noise maps. Moreover, much of the existing research focuses on enhancing image reconstruction quality to improve preservation, resulting in poor editing performance and artifacts in the edited image (e.g., failing to transform a butterfly into a parrot, as shown in Figure 1).

In this paper, we propose Editable Noise Map Inversion (ENM Inversion). Unlike previous approaches that concentrate on reconstructing the source image, ENM Inversion identifies optimal noise maps tailored for both the original and edited images. To achieve this objective, we utilize an editable noise refinement that searches for the ideal noise maps, which are optimized for both preservation and editability, as shown in Figure 2(a). Our approach leverages the observation that high-quality edits can be effectively achieved by aligning reconstructed and edited noise maps in each inversion step. By reducing the difference between two versions of noise vectors, we can effectively imprint the desired image onto the noise and improve editability. Additionally, to maintain the content of source image, we decrease reconstruction errors in the inverted noise maps. This ensures that ENM Inversion achieves an ideal balance between preserving the details of the original image and edit fidelity.

Our proposed approach can be applied to attention-based image editing pipelines, such as Prompt-to-Prompt (Hertz et al., 2022), MasaCtrl (Cao et al., 2023), and Plug-and-Play (Tumanyan et al., 2023). Experimental results demonstrate that ENM Inversion outperforms existing methods across diverse image editing tasks both quantitatively and qualitatively. In addition to image editing, ENM Inversion can be extended to video editing, enabling manipulation of visual content across frames.

## 2. Related Work

As shown in Figure 2, diffusion-based editing pipelines typically consist of two processes: an inversion process, which transforms the input image into Gaussian noise, and an editing process, which commonly employs attention control to modify the image.

**Editing with Diffusion Models.** Text-to-image diffusion models such as Stable Diffusion (Rombach et al., 2022), Imagen (Saharia et al., 2022), and DALL·E2 (Ramesh et al., 2022) have significantly advanced the field of image generation. Leveraging these powerful models, current methods (Meng et al., 2021; Hertz et al., 2022; Patashnik et al., 2023; Park et al., 2024; Hertz et al., 2024) apply their capabilities to text-guided image editing without additional training. Prompt-to-Prompt (P2P) (Hertz et al., 2022) was the first to achieve remarkable localized modifications by manipulating cross-attention maps in Stable Diffusion. As illustrated in Figure 2(b), P2P replaces the attention maps of the editing path with the corresponding maps of the reconstruction path. Plug-and-Play (PNP) (Tumanyan et al., 2023) is another approach that modifies spatial features and self-attention maps to control the structure of the generated image. Pix2pix-Zero (Parmar et al., 2023) utilizes cross-attention guidance for zero-shot image-to-image translation. MasaCtrl (Cao et al., 2023) converts the self-attention in diffusion into mutual self-attention, enabling non-rigid edits. Recent studies (Wu et al., 2023; Wang et al., 2023; Liu et al., 2024) have adopted this technique for video editing, which aims to modify the source video contents according to text prompts. When applied to real-world images, these editing methods require inverting the images into the noise vectors.

**Inversion with Editing Methods.** Inversion techniques are broadly categorized into DDIM-based and DDPM-based methods. DDIM inversion is a widely adopted method that estimates the noise map to reconstruct the original image through deterministic sampling. However, when Classifier-Free Guidance is applied, this method incurs significant errors, leading to poor reconstruction. Optimization-based approaches, such as Null-Text Inversion (NTI) (Mokady et al., 2023), and Negative-prompt Inversion (NPI) (Miyake et al., 2023) adjust text embedding to achieve accurate reconstruction. Methods like EDICT (Wallace et al., 2023; Zhang et al., 2025) offer an exact inversion via an auxiliary neural network. PNP Inversion (Ju et al., 2024) more efficiently preserves the essential content in the source image by disentangling the inversion process into distinct source and target branches. AIDI (Pan et al., 2023), FPI (Meiri et al., 2023) and ReNoise (Garibi et al., 2024) utilize fixed-point iteration processes at each inversion step to obtain accurate noise maps. While these methods achieve high-quality reconstruction, they often exhibit reduced editability. Instead of a deterministic method, DDPM inversion (Huberman-Spiegelglas et al., 2024), often exhibits instability in editing, particularly in content preservation and edit fidelity due to the stochastic nature of the process.

Since inverted noise maps are designed to reconstruct the source image, existing methods often face challenges in achieving high fidelity to both the original content and the desired edits. Unlike previous approaches, ENM inversion aims to encode the target image into noise maps. This enables our method to achieve content consistency and precise editing. Furthermore, by integrating our inversion with existing attention-based editing pipelines, such as P2P, MasaCtrl, we demonstrate a significant improvement in both content preservation and edit fidelity.

## 3. Method

### 3.1. Preliminaries

**Text-guided Diffusion Model.** Diffusion models are generative models that progressively add Gaussian noise to data through a forward process and denoise it through a reverse process. The forward process can be formulated as:

$$z_t = \sqrt{\alpha_t} z_0 + \sqrt{1 - \alpha_t}\epsilon, \quad \epsilon \sim \mathcal{N}(0, I), \qquad (1)$$

where $z_0$ represents the original data, $z_t$ is the noisy data at timestep $t$, and $\{\alpha_t\}_{t=0}^{T}$ is a predefined noise schedule for $t \in [1, T]$.

The training objective of diffusion models involves learning a noise prediction network, $\epsilon_\theta$, parameterized by a neural network. The network is conditioned on embedding $C$, which is obtained the given text $P$ and is optimized to minimize the loss:

$$\mathcal{L}_{dm} = \mathbb{E}_{z_0, \epsilon, t}\left[\|\epsilon - \epsilon_\theta(z_t, t, C)\|^2\right]. \qquad (2)$$

**DDIM Inversion.** DDIM inversion is useful for real image editing as it allows encoding an image into the latent space of the diffusion model. DDIM employs a deterministic reverse process to generate an image from $z_T$:

$$z_{t-1} = \frac{\sqrt{\alpha_{t-1}}}{\sqrt{\alpha_t}} z_t + \qquad (3)$$

$$\sqrt{\alpha_{t-1}}\left(\sqrt{\frac{1}{\alpha_{t-1}} - 1} - \sqrt{\frac{1}{\alpha_t} - 1}\right)\epsilon_\theta(z_t, t, C).$$

This process can be written as $z_{t-1} \leftarrow f(z_t, t, C)$. Based on the assumption that the ODE formula can be reversed in the limit of infinitesimally small steps, DDIM inversion can be derived from Equation (3):

$$z_{t+1} = \frac{\sqrt{\alpha_{t+1}}}{\sqrt{\alpha_t}} z_t + \qquad (4)$$

$$\sqrt{\alpha_{t+1}} \left( \sqrt{\frac{1}{\alpha_{t+1}} - 1} - \sqrt{\frac{1}{\alpha_t} - 1} \right) \epsilon_\theta(z_t, t, C).$$

We can summarize the inversion process as $z_t \leftarrow f_{inv}(z_{t-1}, t-1, C)$. DDIM inversion introduces approximation errors at each timestep, leading to a significant decrease in both reconstruction accuracy and editability. To minimize these errors, fixed-point iteration methods compute the ideal noise for the source image by solving implicit functions. This approach improves reconstruction quality but exhibits limited editability.

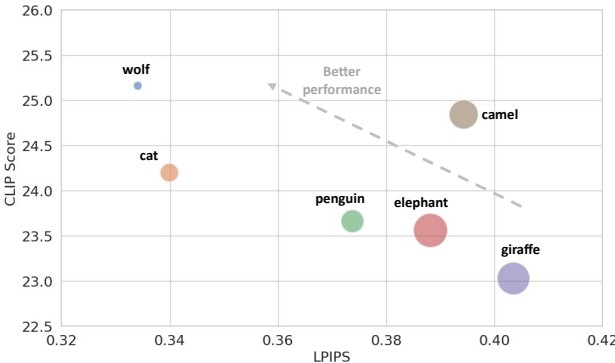

*Figure 3.* Relationship Between Editing Performance and Noise Map Differences. Editing performance is evaluated using LPIPS, which measures perceptual similarity, and CLIP score, which assesses alignment with the target prompt. The size of each circle indicates the magnitude of differences between the reconstructed and edited noise maps at the 30th inversion step. Smaller noise map differences correlate with better editing performance.

### 3.2. Editable Noise Map Inversion

Editable Noise Map Inversion (ENM Inversion) proposes a method for inverting noise maps that are suitable for both reconstruction and editing. The primary observation motivating our approach is that noise maps, which enable high-quality edits, exhibit minimal differences between those reconstructed with the source prompt and those edited with the target prompt. Our analysis is inspired by Pix2pix-Zero (Parmar et al., 2023), which computes editing directions by using changes in attention maps between reconstructed and edited noisy inputs. Unlike attention maps, which mainly reflect high-level semantic relationships, noise maps inherently capture low-level structural details and spatial context. Therefore, we analyze the differences between reconstructed and edited noise maps to improve the inversion process.

---

**Algorithm 1** Editable Noise Map Inversion

> **Input:** A source image $z_0$, number of inversion steps $T$, source prompt $P_{src}$, target prompt $P_{tgt}$, number of refinement steps $\mathcal{K}$, threshold $\tau$
> **Output:** Inverted noise maps $\{z_T, \ldots, z_1\}$.
> **for** $t \leftarrow 1, 2, \ldots, T$ **do**
> $\quad z_t \leftarrow f_{inv}(z_{t-1}, t-1, C_{src})$
> $\quad z_{t-1}^e \leftarrow f(z_t, t, C_{tgt})$
> $\quad$ **for** $i \leftarrow 1, \ldots, \mathcal{K}$ **do**
> $\quad\quad z_{t-1}^r \leftarrow f(z_t, t, C_{src})$
> $\quad\quad L_{edit} = \|z_{t-1}^e - z_{t-1}^r\|_2$
> $\quad\quad L_{recon} = \|z_{t-1} - z_{t-1}^r\|_2$
> $\quad\quad z_t \leftarrow z_t - \nabla(L_{recon} + \lambda L_{edit})$
> $\quad\quad$ **if** $L_{recon} + \lambda L_{edit} < \tau$ **then**
> $\quad\quad\quad$ Break
> $\quad\quad$ **end if**
> $\quad$ **end for**
> **end for**

---

To investigate the properties of noise maps that lead to effective manipulation, we analyze cases where editing has succeeded and failed. Using AFHQ images (Choi et al., 2020), we perform various edits where the source image of a dog is transformed into different target objects (e.g., a cat, a penguin). We focus on the differences between the reconstructed noise map and the edited ones at the 30th inversion step, evaluating their impact on editing performance. As shown in Figure 3, we observe that smaller differences between the reconstructed and edited noise maps have strongly correlated with better editing performance. Similar trends are observed across other inversion steps, and a detailed analysis of these is provided in Appendix B.

By leveraging these observations, we implement an editable noise refinement. Specifically, we align the inversion direction towards the ideal noise at each inversion step by reducing the difference between the reconstructed and the edited noise maps:

$$L_{edit} = \|f(z_t, t, C_{src}) - f(z_t, t, C_{tgt})\|_2 \qquad (5)$$

where $C_{src}$ and $C_{tgt}$ represent the source and target text embedding, respectively.

In addition, to prevent the noise maps from deviating excessively from the source image, ENM Inversion incorporates an additional term to reduce the reconstruction error $L_{prev} = \|z_{t-1} - f(z_t, t, C_{src})\|_2$. To improve editability and content preservation, our method iteratively updates $z_t$ by minimizing the following loss function:

$$\underset{z_t}{\arg\min}\, L = L_{prev} + \lambda L_{edit} \qquad (6)$$

where, $\lambda$ is a hyperparameter weighting factor for the edit-

*Table 1.* Quantitative comparisons of inversion methods across text-guided image editing techniques: Prompt-to-Prompt (P2P) (Hertz et al., 2022), MasaCtrl (Cao et al., 2023), and Plug-and-Play (PnP) (Tumanyan et al., 2023), on the PIE-Bench dataset. Our method consistently outperforms other baseline methods in editing performance. Best results are highlighted in **bold**, and second-best results in underline.

| Method | | Structure | Background Preservation | | | | CLIP Similariy | |
|---|---|---|---|---|---|---|---|---|
| Inverse | Editing | Distance$_{\times 10^3}$ ↓ | PSNR ↑ | LPIPS$_{\times 10^3}$ ↓ | MSE$_{\times 10^4}$ ↓ | SSIM$_{\times 10^2}$ ↑ | Whole ↑ | Edited ↑ |
| **DDIM** | **P2P** | 69.43 | 17.87 | 208.80 | 219.88 | 71.14 | 25.01 | **22.44** |
| **NTI** | **P2P** | 13.44 | 27.03 | 60.67 | 35.86 | 84.11 | 24.75 | 21.86 |
| **StyleD** | **P2P** | 11.65 | 26.05 | 66.10 | 38.63 | 83.42 | 24.78 | 21.72 |
| **NMG** | **P2P** | 22.83 | 26.01 | 79.42 | 109.53 | 82.40 | 24.53 | 21.60 |
| **EF** | **P2P** | 18.05 | 24.55 | 91.88 | 94.58 | 81.57 | 23.97 | 21.03 |
| **PNPInv** | **P2P** | 11.65 | 27.22 | 54.55 | 32.86 | 84.76 | 25.02 | 22.10 |
| **Ours** | **P2P** | **10.13** | **28.19** | **45.26** | **27.02** | **86.29** | **25.30** | 22.12 |
| **DDIM** | **MasaCtrl** | 28.38 | 22.17 | 106.62 | 86.97 | 79.67 | 23.96 | 21.16 |
| **NMG** | **MasaCtrl** | 38.72 | 20.33 | 127.21 | 135.03 | 77.48 | 24.54 | 21.32 |
| **PNPInv** | **MasaCtrl** | 24.70 | 22.64 | 87.94 | 81.09 | 81.33 | 24.38 | 21.35 |
| **Ours** | **MasaCtrl** | **22.89** | **23.01** | **83.99** | **77.55** | **82.34** | **24.62** | **21.44** |
| **DDIM** | **PnP***[*] | 28.22 | 22.28 | 113.46 | 83.64 | 79.05 | 25.41 | 22.55 |
| **PNPInv** | **PnP***[*] | 24.29 | 22.46 | 106.06 | 80.45 | 79.68 | 25.41 | 22.62 |
| **Ours** | **PnP***[*] | **18.44** | **25.32** | **78.53** | **46.34** | **83.57** | **25.57** | **22.63** |

[*] use Stable Diffusion v1.5 as base model (others all use Stable Diffusion v1.4)

ing alignment term. Furthermore, for efficient inversion, we introduce a pre-defined threshold $\tau$ at each timestep $t$. By iteratively refining $z_t$ to minimize this loss, ENM Inversion imprints the target image more strongly onto the noise maps while preserving content from the source image. We summarize our proposed process in Algorithm 1.

### 3.3. Extension to Video Editing

Video editing focuses on modifying source video content according to a given text prompt. Recent methods such as Tune-A-Video (TAV) (Wu et al., 2023) and Video-P2P (Liu et al., 2024) have extended text-to-image diffusion models to the video domain, enabling text-guided video editing. However, these approaches treat video frames as a whole for editing, which often results in poor editability and temporal inconsistency between frames.

To overcome these limitations of video editing, we integrate our inversion into Video-P2P. First, we fine-tune an image diffusion model for text-to-video modeling. Subsequently, we utilize ENM Inversion to invert each frame of the source video into noise maps that preserve the structure of the original video and provide enhanced editability. Finally, we apply attention control across all frames to ensure consistent modifications throughout the video. This approach addresses limitations in previous methods by enabling high-quality edits while maintaining structural and

temporal coherence between frames.

## 4. Experiments

### 4.1. Experiments Setup

**Datasets.** To evaluate the effectiveness and efficiency of our proposed ENM Inversion, we use two datasets. For image editing, we use the PIE-Bench introduced by (Ju et al., 2024), which contains high-resolution images, each accompanied by 9 distinct editing tasks, covering a diverse range of semantic and structural modifications. For video editing, we follow previous works (Bar-Tal et al., 2022; Liu et al., 2024) and use videos from the DAVIS dataset (Pont-Tuset et al., 2017) and the Internet to evaluate our approach. The two datasets consist of paired source and target prompts, along with target region masks for localized evaluation.

**Comparison Methods.** We quantitatively and qualitatively compare ENM Inversion with existing inversion techniques. These include DDIM Inversion (Song et al., 2020), Null-Text Inversion (NTI) (Mokady et al., 2023), StyleDiffusion (StyleD) (Li et al., 2023), Noise Map Guidance (NMG) (Cho et al., 2024), Edit-Friendly DDPM Inversion (EF) (Huberman-Spiegelglas et al., 2024), and PnP Inversion (PNPInv) (Ju et al., 2024). For image editing, these inversion methods are integrated with three text-guided image editing techniques: Prompt-to-Prompt (P2P) (Hertz et al.,

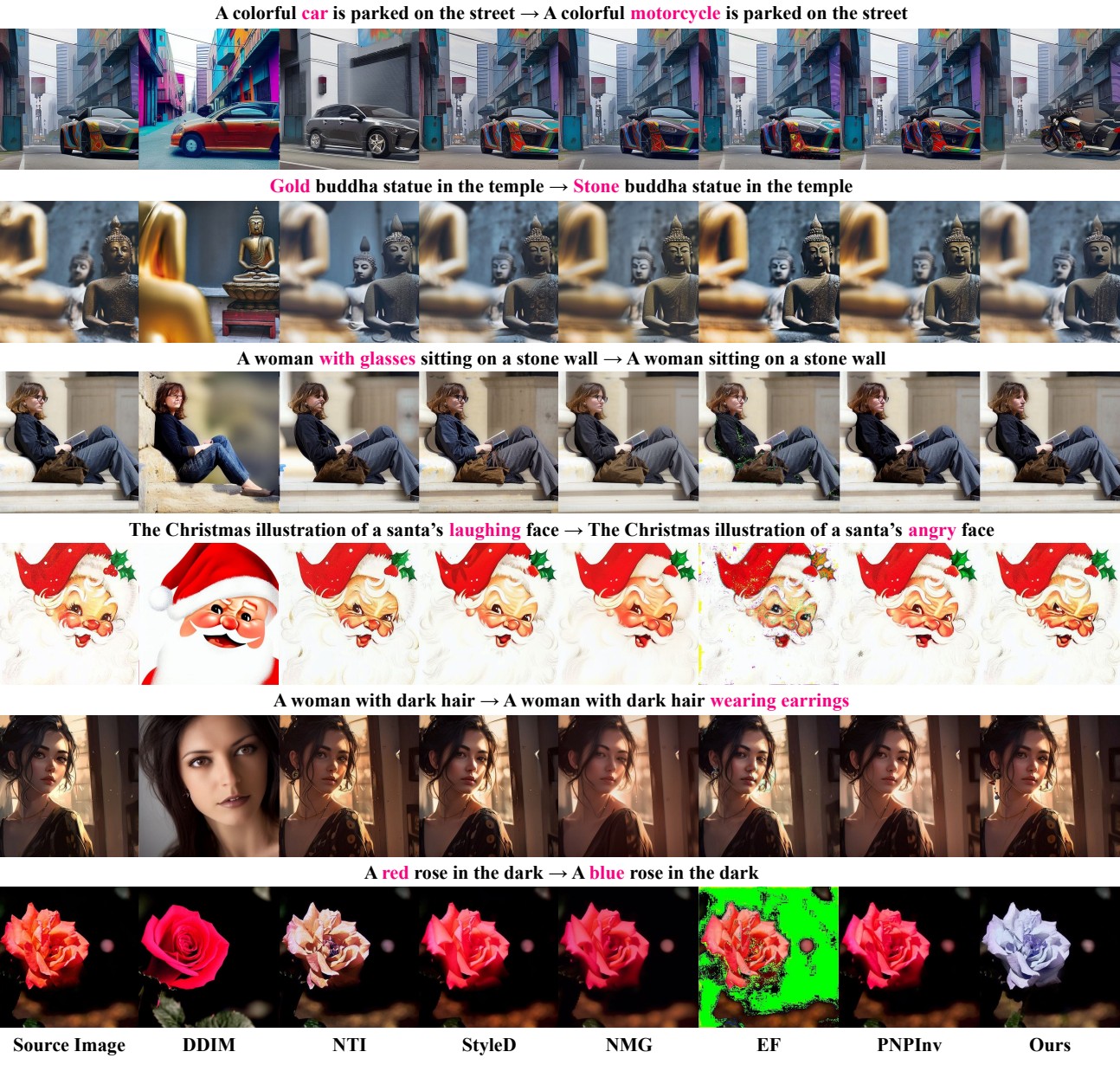

*Figure 4.* Qualitative comparisons of various inversion methods using Prompt-to-Prompt (P2P) (Hertz et al., 2022). Other inversion techniques result in an inconsistent background or structure with the source image or exhibit limited editing capabilities. Our approach not only retains high fidelity to the source image but also demonstrates superior editing capabilities.

2022), MasaCtrl (Cao et al., 2023), and Plug-and-Play (PnP) (Tumanyan et al., 2023). For video editing, DDIM Inversion and NTI are applied to Tune-A-Video (TAV) (Wu et al., 2023) and Video-P2P (Liu et al., 2024), respectively.

**Evaluation Metrics.** To quantitatively assess the performance of our method across various aspects, we utilize the metrics established by PNPInv. We measure structure distance using the DINO score (Tumanyan et al., 2022), and assess background preservation through PSNR, LPIPS (Zhang et al., 2018), MSE, and SSIM (Wang et al., 2004). Background preservation is specifically computed in the unedited regions, defined by areas outside the editing mask. Additionally, we evaluate target prompt-image consistency by computing CLIP Similarity (Wu et al., 2021) for the entire image and the regions within the editing mask. For video editing, we further evaluate temporal consistency by incorporating the *Temp* metric (Esser et al., 2023).

**Implementation Details.** We perform all experiments on

**[Input Video] A camel walking in a fenced in area.**

**[Input Video] A brown bear is walking on the beach.**

**[DDIM + TAV] A elephant walking in a fenced in area.**

**[DDIM + TAV] A brown bear is walking on the grass.**

**[NTI + Video-P2P] A elephant walking in a fenced in area.**

**[NTI + Video-P2P] A brown bear is walking on the grass.**

**[Ours + Video-P2P] A elephant walking in a fenced in area.**

**[Ours + Video-P2P] A brown bear is walking on the grass.**

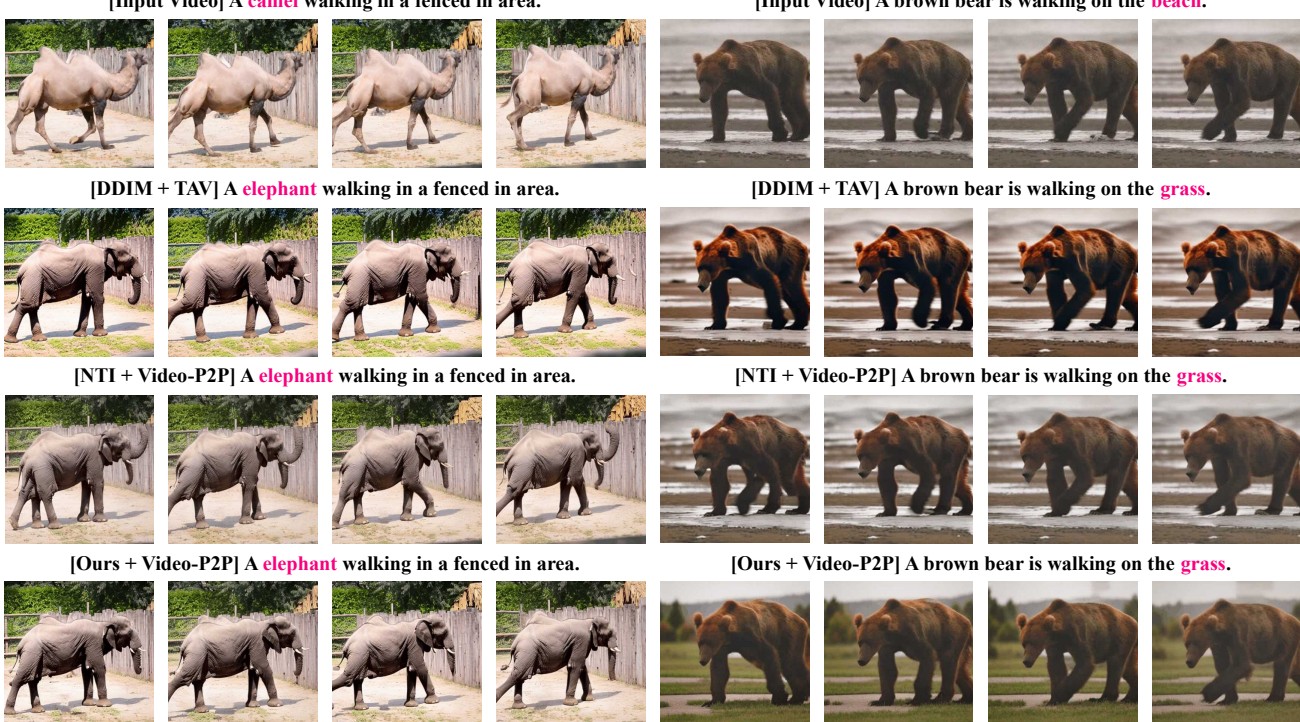

*Figure 5.* Qualitative comparison of our inversion using Video-P2P (Liu et al., 2024) for video editing. Our method demonstrates superior performance in terms of temporal consistency, content maintenance, and editing quality, when modifying backgrounds or objects.

a single RTX3090. Following previous studies (Ju et al., 2024; Liu et al., 2024), we use Stable Diffusion V1-4 for P2P and MasaCtrl, while Stable Diffusion V1-5 is used for PnP, TAV, and Video-P2P. We employ a DDIM schedule with 50 steps and apply Classifier-Free Guidance of 7.5 for the editing process. For editing experiments, we adapt the default parameters for cross-attention injection and self-attention injection.

### 4.2. Comparison with Image Editing

**Quantitative Comparisons.** Table 1 provides the quantitative evaluations of various inversion methods on PIE-Bench. Compared to existing methods, ENM Inversion achieves notable improvements in most metrics. Our approach demonstrates superior performance in text-image alignment and structure preservation, achieving higher CLIP Similarity, DINO Score, PSNR, and SSIM, while maintaining lower LPIPS and MSE values.

We also compare the inference time of different inversion methods integrated with P2P, as shown in Table 2. ENM Inversion achieves better editing performance and is more efficient than NTI and StyleD.

**Qualitative Results.** We present a visual comparison of inversion methods using P2P in Figure 4. Our approach consistently surpasses competing methods in various editing tasks, such as changing content, removing an object, and modifying attributes like material or color. Existing methods struggle with various editing tasks such as changing a car into a motorcycle (the 1st row), removing glasses from a face (the 3rd row), and modifying the color of a rose (the 6th row). NTI faces challenges in maintaining the details of the original image (the 2nd and 3rd rows). Furthermore, both NTI and PnPInv attempt to modify a smiling face to an angry expression but leave residual traces of the smile (the 4th row). EF often introduces noise artifacts into the generated image (the 4th and 6th rows). In contrast, our method excels in preserving structure of the original image while achieving higher alignment with the text prompt compared to existing approaches. For additional qualitative results using other editing methods, please see Appendix F.

*Table 2.* **Inference time of inversion techniques.**

| Method | Time (s) |
| --- | --- |
| DDIM | **18.22** |
| NTI | 148.48 |
| StyleD | 382.98 |
| NMG | 36.48 |
| EF | 19.10 |
| PNPInv | 28.17 |
| Ours | 38.87 |

*Table 3.* Quantitative comparisons of performance between fixed-point iteration methods and our approach on the PIE-Bench dataset. Following their settings, AIDI and FPI are combined P2P, while ReNoise utilizes the img2img pipeline of Stable Diffusion V1-4.

| Method | | Structure | Background Preservation | | | | CLIP Similariy | |
|---|---|---|---|---|---|---|---|---|
| **Inverse** | **Editing** | **Distance**$_{\times 10^3}$ ↓ | **PSNR** ↑ | **LPIPS**$_{\times 10^3}$ ↓ | **MSE**$_{\times 10^4}$ ↓ | **SSIM**$_{\times 10^2}$ ↑ | **Whole** ↑ | **Edited** ↑ |
| **DDIM** | **P2P** | 69.43 | 17.87 | 208.80 | 219.88 | 71.14 | 25.01 | **22.44** |
| **AIDI** | **P2P** | 12.19 | 26.96 | 57.92 | 39.82 | 84.17 | 24.96 | 22.01 |
| **FPI** | **P2P** | 14.71 | 26.61 | 61.97 | 37.64 | 83.52 | 23.93 | 21.35 |
| **ReNoise** | **/** | 22.60 | 25.19 | 85.29 | 49.51 | 82.30 | 23.78 | 21.15 |
| **Ours** | **P2P** | **10.13** | **28.19** | **45.26** | **27.02** | **86.29** | **25.30** | 22.12 |

*Table 4.* Quantitative evaluation against baselines in video editing. We measure target text alignment (CLIP Score), background preservation (LPIPS and SSIM), and temporal consistency (TEMP).

| Method | CLIP ↑ | LPIPS ↓ | SSIM ↑ | TEMP ↑ |
|---|---|---|---|---|
| **DDIM+TAV** | 26.13 | 169.80 | 68.02 | **0.9464** |
| **NTI+Video-P2P** | 26.14 | 104.64 | 74.74 | 0.9451 |
| **Ours+Video-P2P** | **26.57** | **98.24** | **75.43** | 0.9454 |

### 4.3. Comparison with Video Editing

**Quantitative Comparisons.** We present the quantitative evaluation results in Table 4, which compare the performance of our method with state-of-the-art models. Compared to DDIM+TAV and NTI+Video-P2P, we achieve higher CLIP Score, SSIM, and lower LPIPS, demonstrating better alignment with text, superior structural integrity, and perceptual accuracy. Moreover, our approach produces these results in significantly less time compared to NTI. Table 4 shows that ENM inversion excels in text fidelity and content preservation, delivering high-quality video with improved efficiency.

**Qualitative Results.** Figure 5 presents the results of video editing. DDIM+TAV struggles to preserve the content of the input video accurately. NTI+Video-P2P fail to consistently align with the user-specified prompts, leading to the generation of artifacts and unrealistic outputs. Both baselines also exhibit poor editing performance due to limited editability of the latent noise (right side of Figure 5). In contrast, our inversion with Video-P2P performs temporally consistent video editing, while preserving fine details of the original video and achieving high edit fidelity.

### 4.4. Comparison With Fixed-Point Methods

In Table 3, we compare ENM Inversion with fixed-point iteration methods for image editing. Fixed-point iteration methods (Pan et al., 2023; Meiri et al., 2023; Garibi et al., 2024) improve the structure and background preservation in DDIM Inversion, but clip similarity has decreased. This

result occurs because these methods focus on solving an implicit function to ensure accurate reconstruction of the source image, leading to lower editability. In contrast, ENM Inversion optimizes a loss function that refines the noise map for editing, aligning it with the target image while maintaining details from the source. This allows us to enhance both content preservation and edit performance simultaneously.

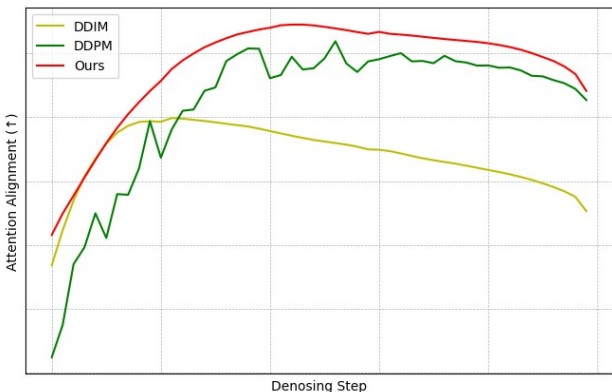

*Figure 6.* Cross-attention alignment for image editing across denoising steps with different inversion techniques.

### 4.5. Exploring the Editability of Noise Maps

Since the editing performance is measured solely based on CLIP Similarity, we add experiments for the editability of noise maps. To assess edit fidelity, we evaluate cross-attention alignment for image editing across denoising steps (Tumanyan et al., 2023; Butt et al., 2025). Specifically, we compute the cross-attention map corresponding to the target text and measure its overlap with the target region mask. The alignment is denoted as score $= \sum(A_t \cdot M)/\sum A_t$. $A_t$ represents the cross-attention map at timestep $t$, and $M$ is the binary mask indicating the edited region. We compare our method against DDIM Inversion and DDPM Inversion, applying P2P in Figure 6. DDIM Inversion initially improves alignment at early timesteps, and gradually declines after a certain point, leading to reduced editability.

DDPM Inversion achieves better performance than DDIM but exhibits instability and low initial alignment due to its stochastic nature. Our method achieves higher scores than both inversion techniques. Furthermore, unlike DDPM Inversion, which suffers from instability, our approach ensures stable improvement in alignment across timesteps. With our editable noise maps, we can effectively generate edited images.

## 5. Conclusion

In this paper, we propose ENM Inversion, an effective inversion technique for high-quality real image editing. By refining noise maps to align with both the source and target images, ENM Inversion encodes the target image more strongly into the noise maps, thus enabling high-quality edits while preserving the details of the source image. Experimental results demonstrate the superiority of our method over existing methods in various image editing tasks. Moreover, with the high edit flexibility of our noise maps, our approach can be easily extended to video editing.

## Acknowledgements

This work was supported by the National Research Foundation of Korea (NRF) grant (No. RS-2025-00520618) and the Institute of Information and communications Technology Planning and evaluation (IITP) grant (No. RS-2020-II201373), funded by the Korean Government (MSIT: Ministry of Science and Information and Communication Technology).

## Impact Statement

This work makes a contribution to the advancement of text-guided image and video editing using diffusion models. From a research perspective, we introduce a new approach to improving editability in diffusion inversion and demonstrate strong performance through extensive experiments. From an industrial perspective, our proposed method holds potential benefits for various practical applications, including creative content generation, personalized media editing, and educational tools that rely on expressive visual modifications.

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

*Table 5.* Analysis on Hyper-parameters of ENM Inversion added with Prompt-to-Prompt (P2P) (Hertz et al., 2022).

| Parameters | Structure | Background Preservation | | | | CLIP Similariy | |
|---|---|---|---|---|---|---|---|
| | Distance$_{\times 10^3}$ ↓ | PSNR ↑ | LPIPS$_{\times 10^3}$ ↓ | MSE$_{\times 10^4}$ ↓ | SSIM$_{\times 10^2}$ ↑ | Whole ↑ | Edited ↑ |
| $\lambda = 5$ | 10.10 | 28.22 | 45.19 | 26.91 | 86.23 | 25.17 | 22.03 |
| $\lambda = 10$ | 10.13 | 28.19 | 45.26 | 27.02 | 86.29 | 25.30 | 22.12 |
| $\lambda = 15$ | 10.38 | 28.13 | 46.33 | 27.25 | 86.15 | 25.32 | 22.07 |
| $\lambda = 20$ | 12.41 | 28.01 | 46.80 | 27.30 | 86.12 | 25.34 | 22.08 |
| $T = 20$ | 8.74 | 28.50 | 44.05 | 25.19 | 86.35 | 24.95 | 21.71 |
| $T = 50$ | 10.13 | 28.19 | 45.26 | 27.02 | 86.29 | 25.30 | 22.12 |
| $T = 75$ | 10.35 | 28.10 | 46.47 | 27.68 | 86.11 | 25.35 | 22.13 |
| $T = 100$ | 10.89 | 28.01 | 48.03 | 27.91 | 86.05 | 25.44 | 22.20 |
| **Default** | **10.13** | **28.19** | **45.26** | **27.02** | **86.29** | **25.30** | **22.12** |

## A. Limitations

Our proposed method has several limitations. First, it relies on the generative capabilities of Stable Diffusion. As a result, if the target image lies outside the domain that Stable Diffusion can generate, our method may fail to edit the image effectively. Another limitation lies in computational efficiency. Unlike existing methods that perform a single inversion per source image and reuse the result across multiple target prompts, our approach requires a separate inversion process for each target text-image combination. This increases the computational cost, especially in scenarios involving multiple edits of the same image. Additionally, the optimizing of noise maps introduces further inference time. While this added cost is relatively small, as shown in Table 2, it may still pose challenges in real-time applications.

## B. Analysis of Noise Map Differences Across Inversion Steps

We analyze the differences between the reconstructed and edited noise maps across various inversion steps to better understand their impact on editing performance. Figure 7 (left) provides an analysis of noise map differences across inversion steps for various editing cases. For editing cases with lower performance (e.g., transforming a dog into an elephant or a giraffe), we observe larger gaps across inversion steps. This indicates that greater deviations between the reconstructed and edited noise maps can lead to poorer editing outcomes. Figure 7 (right) illustrates the relationship between editing performance and noise map changes at the 30th inversion step. Compared to DDIM inversion, our approach achieves a smaller gap between the reconstructed and edited noise maps, which leads to higher editing performance.

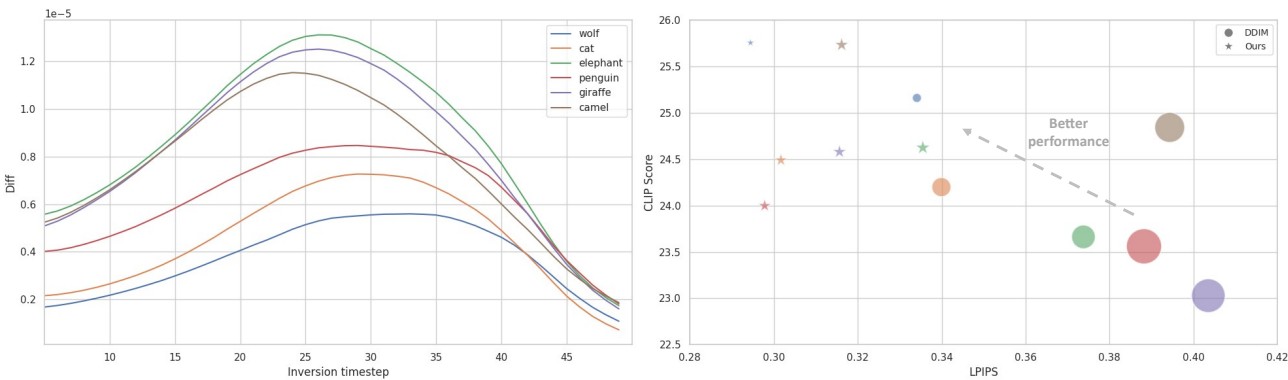

*Figure 7.* Noise map differences across inversion steps for various editing cases (left). Relationship between editing performance and noise map differences at the 30th inversion step (right). The size of stars and circles represents the magnitude of the difference.

*Table 6.* Quantitative comparisons of performance between flow-based models and our approach on the PIE-Bench dataset. Our method outperforms other flow-based methods in editing performance.

| Method | Structure | Background Preservation | | | | CLIP Similariy | |
|---|---|---|---|---|---|---|---|
| | Distance$_{\times 10^3}$ ↓ | PSNR ↑ | LPIPS$_{\times 10^3}$ ↓ | MSE$_{\times 10^4}$ ↓ | SSIM$_{\times 10^2}$ ↑ | Whole ↑ | Edited ↑ |
| SDEdit-Flux | 118.97 | 14.41 | 329.92 | 450.06 | 60.82 | **25.06** | 22.50 |
| RFInv | 60.08 | 18.24 | 232.88 | 210.02 | 64.78 | 24.94 | **22.65** |
| Ours + RFInv | **46.38** | **19.77** | **185.86** | **153.90** | **69.57** | 25.05 | **22.65** |

*Table 7.* Comparison of reconstruction quality when using the source prompt $C_{src}$ and a null text during inversion.

| Method | Structure | Background Preservation | | | |
|---|---|---|---|---|---|
| | Distance$_{\times 10^3}$ ↓ | PSNR ↑ | LPIPS$_{\times 10^3}$ ↓ | MSE$_{\times 10^4}$ ↓ | SSIM$_{\times 10^2}$ ↑ |
| $C_{src}$ | 10.13 | 28.19 | 45.26 | 27.02 | 86.29 |
| Null | 9.89 | 28.18 | 46.28 | 27.13 | 86.11 |

## C. Analysis of Hyper-parameters

In Table 5, we provide further experiments to analyze the impact of different choices in our method. As the editing alignment weight $\lambda$ increases, the performance in preserving the structure and background of the image decreases, while the CLIP similarity improves. This trend indicates that a larger $\lambda$ amplifies the editability, increasing the likelihood of edits but also the probability of affecting unintended regions of the image. Additionally, increasing $\lambda$ requires more inference time. To maintain a balanced performance without excessive computational overhead, we select $\lambda = 10$ as a practical choice.

Furthermore, we conducted experiments with DDIM sampling steps $T$ set to 20, 50, 75, and 100 in ENM Inversion. Fewer steps resulted in better preservation of background and structure, while higher step counts improved CLIP similarity. To achieve balanced performance across all metrics, we set $T = 50$ as the default setting.

## D. Comparison with Flow-Based Models

While diffusion-based methods have demonstrated strong capabilities in reconstructing and editing images, recent advances in flow-based models offer an alternative and promising direction. In particular, Rectified Flow (RF) models (Liu et al., 2022; Albergo & Vanden-Eijnden, 2022; Esser et al., 2024) have emerged as an efficient generative framework that leverages reverse Ordinary Differential Equations (ODEs) instead of the Stochastic Differential Equations (SDEs) commonly used in diffusion models. Recent works have explored inversion techniques tailored for Rectified Flows, introducing algorithms such as RF Inversion (RFInv) (Rout et al., 2024). Our method can be integrated into these methods. We conducted additional experiments using the following baselines: SDEdit (Meng et al., 2021)-Flux, RFInv, and Ours + RFInv. As shown in Table 6, our method significantly improves reconstruction quality while preserving the edit fidelity inherent in flow-based models.

## E. Robustness to Different Source Prompts

In our method, accurate image reconstruction during inversion is achieved by minimizing $L_{prev}$ as defined in Equation (6). This loss is computed using the denoising function $f(z_t, t, C_{src})$, where $C_{src}$ is the source prompt provided during inversion. To validate the robustness with respect to a different choice of $C_{src}$, we conduct an additional experiment by setting $C_{src}$ to a null text and comparing the quality against the case where the actual source prompt is used. Table 7 show that there is no significant difference in reconstruction quality between the two settings.

## F. Additional Qualitative Results

We present additional qualitative results using the PIE-Bench dataset. Figure 8, Figure 9, and Figure 10 show comparisons of inversion methods combined with Prompt-to-Prompt, MasaCtrl, and Plug-and-Play techniques. Furthermore, Figure 11

provides additional visualizations for video editing.

**A view of the mountains covered in snow → A view of the mountains covered in leaves**

**A woman in a jacket standing in the rain → A woman in a blouse standing in the rain**

**A black raven sits on a tree stump in the rain → A white raven sits on a tree stump in the rain**

**A long haired cat looking up at something → A short haired cat looking up at something**

**A digital art woman with curly hair standing in front of buildings → A digital art woman with straight hair standing in front of buildings**

**A watercolor illustration of a chicken on a white background → An anime illustration of a chicken on a white background**

| Source Image | DDIM | NTI | StyleD | NMG | EF | PNPInv | Ours |

*Figure 8.* Qualitative results of different inversion methods with Prompt-to-Prompt (P2P).

**An illustration of a cat sitting on top of rock → An illustration of a bear sitting on top of rock**

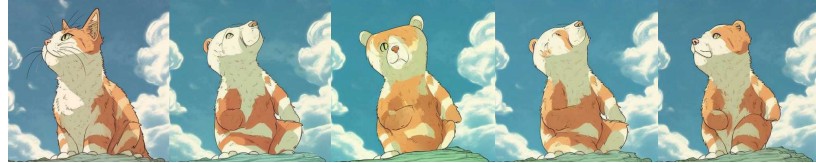

**A bee flies over a flowering tree branch → A flowering tree branch**

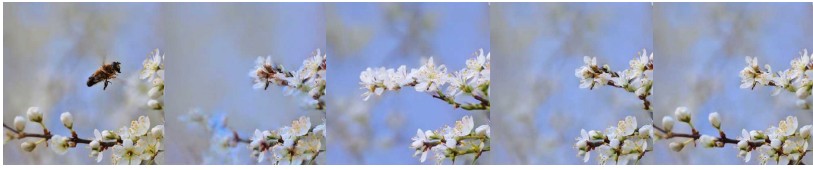

**A woman in a hat and dress walking down … → A woman in a hat and dress running down …**

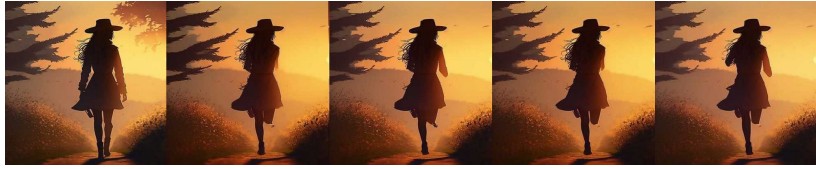

**A serious man → A angry man**

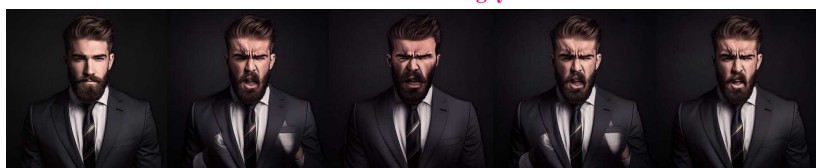

**A lone tree is reflected in the water at night with a bright moon → A lone tree is reflected in the water at night**

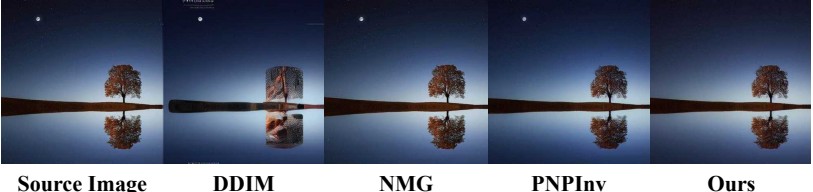

|  Source Image  |  DDIM  |  NMG  |  PNPInv  |  Ours  |

*Figure 9.* Qualitative results of different inversion methods with MasaCtrl.

**A cat sitting in the grass → A cat sitting in the rocks**

**A puppy is sitting in a field of dandelions → A puppy is sitting in a field**

**A man wearing a tie → A man wearing a black and yellow stripes tie**

**A young girl … → Black and white sketch of a young girl …**

**White flowers on a tree … → An oil painting of white flowers on a tree …**

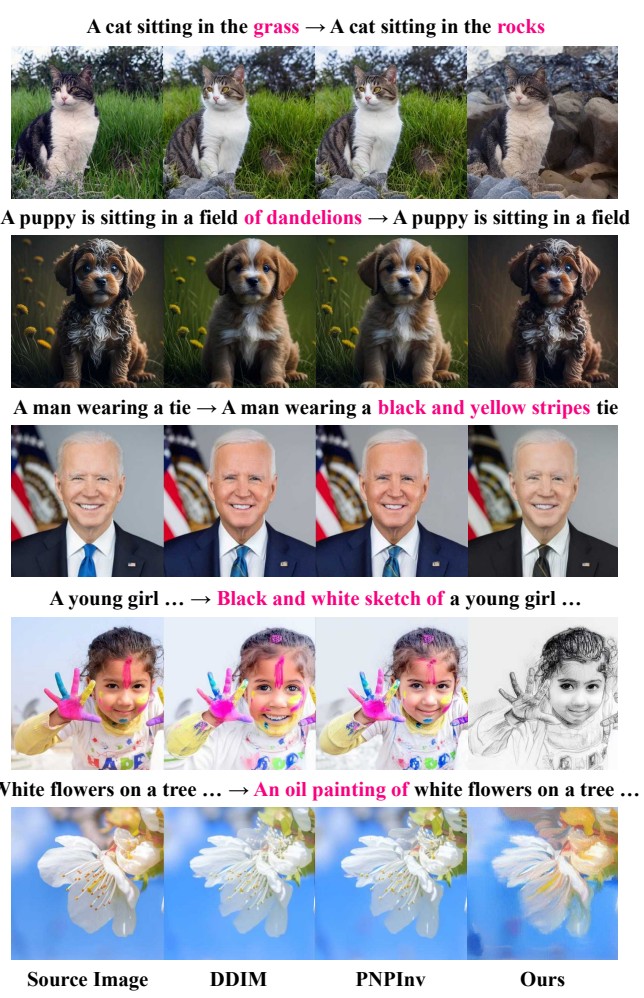

| Source Image | DDIM | PNPInv | Ours |
|---|---|---|---|

*Figure 10.* Qualitative results of different inversion methods with Plug-and-Play (PnP).

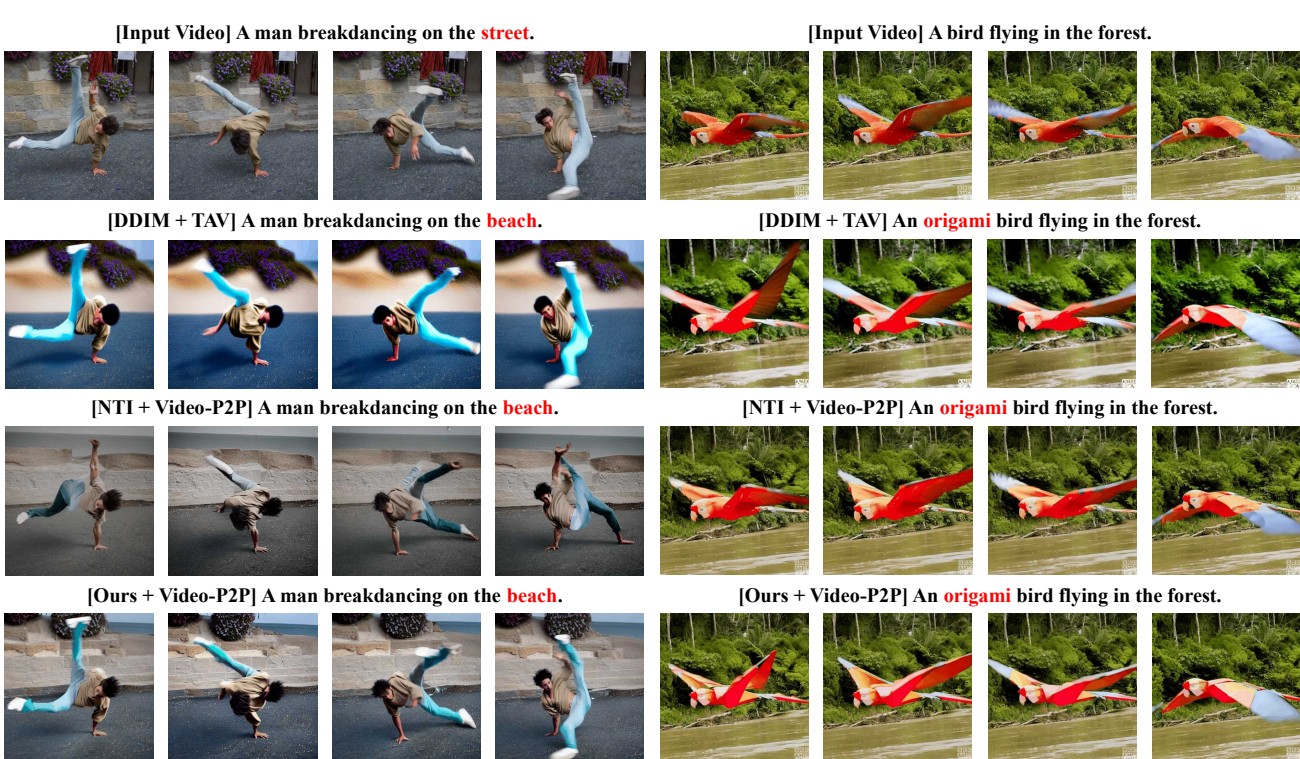

*Figure 11.* Qualitative results of different inversion methods with Plug-and-Play (PnP).

