# OpenReview forum: "Editable Noise Map Inversion: Encoding Target-image into Noise For High-Fidelity Image Manipulation"
_ICML.cc/2025/Conference — ICML 2025 poster_

### Official Review · Reviewer_nzEq · 2025-03-13

**Overall Recommendation:** 4

**Summary:**

This paper proposed a new inversion-based imag/video editing method called ENM inversion. The motivation is to improve text alignment with the target text prompt. The authors proposed editable noise refinement, which conduct inference time optimzation on the intermediate latents. The proposed ressults achieved sota performance on both image and video editing datasets.


## update after rebuttal
As mentioned the in rebuttal reponse, I increase the score to accept only if the authors add related details to the final version. The authors haven't confirm yet, which I take as acknowledgement.

**Claims And Evidence:**

Most claims for valid and supported by evidence. But since the paper claims to be a general image manipulation methods, the authors should test on more types of manipulations. All the results presented in the paper seems to be minor editings like texture, color, style, expression. I'd like to see if it works for more editing tasks: (1) adding object, e.g., adding a hat. (2) multi object editing: e.g., you have a blue toy holding a yellow flower, you want to change to a yellow toy holding a blue flower.

**Essential References Not Discussed:**

There are many papers on inversion-based editing on the rectified-flow models, would be good to have them for completeness.
But not very essential since this paper only compares diffusion models.

**Experimental Designs Or Analyses:**

Question:
1. I think the editability mainly comes from the editing module like PnP, while the proposed method makes the output more consistent to the source image as  eq. (6). So I would expecting that you outperforms the other method mainly on the editing part but not editing part. But it seems from the results that other methods with the same editing module still performs worse (e.g., fig.4 last row only the proposed method is able to change the color). Can you explain?
2. Does the method works on latest flow models like SD3.5/Flux?
3. The recent RF-inversion has similar ideas behind the scene, which construct a new vector field to update intermediate latent at each timestep, would be good to compare with this model or it's variants.

**Methods And Evaluation Criteria:**

The methods are somewhat novel, though there are other works explring updating the latent space, but most of them are in the reverse process, not inverse process. The eval matrics are good. Would be better to add human eval since the metrics are sometimes not very reflective in human preference as noted in many papers like dreambooth and RB-modulation.

Question:
1. In the construction of the loss (5)(6), the assumption is that the src prompt can give you a latents that reconstruct the image well. Can you prove if this assumption is correct? Will different src prompts lead to different results? e.g. "a walking tiger", "a tiger is waling on the ground", "a tiger", "a tiger in the jungle" might all refers to the same src image.

**Other Comments Or Suggestions:**

no.

**Other Strengths And Weaknesses:**

Discussed above.

**Questions For Authors:**

included above.

**Relation To Broader Scientific Literature:**

Mostly comprehensive, added one more suggestions in the experiment section above.

**Theoretical Claims:**

The formulations are correct.

---

> ### Author Rebuttal · Authors · 2025-03-29
>
> We sincerely appreciate you taking the time to review our research. Below, we have provided responses to points raised.
>
> **Claims And Evidence:**
> > I'd like to see if it works for more editing tasks: (1) adding object, e.g., adding a hat. (2) multi object editing: e.g., you have a blue toy holding a yellow flower, you want to change to a yellow toy holding a blue flower.
>
> **Answer:**
> Thank you for your valuable feedback. We have incorporated qualitative results demonstrating the ability of our method to handle object addition. However, our work does not specifically aim to address multi-object editing, and the dataset we used, PIE-Bench, does not contain multi-object editing tasks. We acknowledge the importance of this direction and consider it a promising avenue for future research.
>
> **Methods And Evaluation Criteria Question:**
> > In the construction of the loss (5)(6), the assumption is that the src prompt can give you a latents that reconstruct the image well. Can you prove if this assumption is correct? Will different src prompts lead to different results?
>
> **Answer:**
> While different source prompts can lead to slight variations in the results, the differences are not significant. As mentioned in NTI, inversion using a null source prompt already provides a latent that reconstructs the image well. When a source prompt is provided, even with the Classifier-Free Guidance scale set to 1, the reconstruction quality remains high. Additionally, NTI demonstrates that optimizing only the text embedding (rather than the latent) is sufficient for accurate image reconstruction. In our approach, we follow the same setup as other comparison methods by setting the scale to 1 during inversion.
>
> **Experimental Designs Or Analyses:**
> > I think the editability mainly comes from the editing module like PnP, while the proposed method makes the output more consistent to the source image as eq. (6). So I would expecting that you outperforms the other method mainly on the editing part but not editing part. But it seems from the results that other methods with the same editing module still performs worse (e.g., fig.4 last row only the proposed method is able to change the color). Can you explain?
>
> **Answer:**
> Thank you for your insightful review. You raise a valid point regarding the role of the editing module in determining editability. However, editability is influenced not only by the editing module but also by the initial noise. As explained in [1], the choice of latent space plays a important role in generating specific concepts. Our method is designed not only to better preserve the source image but also to find a latent that facilitates the generation of the edited image. Furthermore, Figure 6 demonstrates that our inversion approach allows the attention map to be applied to the edited region more quickly and stably. This indicates that our latent generates images more quickly and consistently with the source image while also enabling stronger enforcement of edits compared to other inversion methods.
>
> > Does the method works on latest flow models like SD3.5/Flux?
>
> > The recent RF-inversion has similar ideas behind the scene, which construct a new vector field to update intermediate latent at each timestep, would be good to compare with this model or it's variants.
>
> **Answer:**
> Yes, our method is applicable to Flux and other flow-based models. We have conducted additional experiments specifically on flow-based models [2][3] and integrated our approach with RF-inversion [3] to evaluate its effectiveness.
>
> | **Method** | **Structure Distance** ↓ | **PSNR** ↑ | **LPIPS** ↓ | **MSE** ↓ | **SSIM** ↑ | **CLIP Similarity (Whole)** ↑ | **CLIP Similarity (Edited)** ↑ |
> |--------------|----------------------------|------------|------------------|------------------|------------------|-------------------------|-------------------------|
> | SDEdit-Flux [2]  | 118.97                        | 14.41      | 329.92            | 450.06            | 60.82            | 25.06                   | 22.50                   |
> | RFInv [3]  | 60.08                        | 18.24      | 232.88            | 210.02            | 64.78            | 24.94                   | 22.65                   |
> | Ours + RFInv  | 46.38                        | 19.77      | 185.86            | 153.90            | 69.57            | 25.05                   | 22.65                   |
>
> Table 1. Results of comparing and combining our method with flow-based models.
>
> As shown in the table, our method enhances the performance of RF-Inversion, demonstrating its capability to work effectively on models like Flux.
>
> We appreciate your insightful comments and suggestions.
>
> **Reference**
>
> [1] Generating images of rare concepts using pre-trained diffusion models
>
> [2] SDEdit: Guided Image Synthesis and Editing with Stochastic Differential Equations
>
> [3] Semantic Image Inversion and Editing using Rectified Stochastic Differential Equations

---

> > ### Comment · Reviewer_nzEq · 2025-04-06
> >
> > I appraciate the authors for the response. I will increase the score to accept on condition that the authors incorpoate the following experiments in the finla version. (1) adding objects (2) results to demonstrate the authors statement "While different source prompts can lead to slight variations in the results, the differences are not significant." (3) The other results in the authors response above. Thanks.

---

### Official Review · Reviewer_Qit8 · 2025-03-14

**Overall Recommendation:** 3

**Summary:**

This paper propose ENM Inversion, a technique for high-quality real image editing. By refining noise maps to align with both the source and target images, ENM Inversion encodes the target image more effectively into the noise maps, allowing for high-quality edits while preserving the source image's details.

## update after rebuttal
The authors' rebuttal has addressed my previous concerns. After considering the feedback from the other reviewers and the authors' response, I have decided to maintain my original evaluation to a Weak Accept.

**Claims And Evidence:**

Yes.

**Essential References Not Discussed:**

None

**Experimental Designs Or Analyses:**

I reviewed the experimental results presented in Tables 1, 2, 3 and 4, and they appear to be correct.

**Methods And Evaluation Criteria:**

Yes.

**Other Comments Or Suggestions:**

None

**Other Strengths And Weaknesses:**

**Paper Strengths:**

The paper is well written. The main motivation is clear and easy to understand.

**Paper Weaknesses:**

1. What does $Z_t^{s}$ represent in Figure 2? There is no clear definition of this symbol.
2. The authors state that "smaller differences between the reconstructed and edited noise maps are strongly correlated with better editing performance." However, this is not always the case. As shown in Figure 3, the camel, elephant, and giraffe exhibit similar noise map differences, yet their editing performance varies. Have the authors attempted using the same editing prompt, such as "camel," to perform different types of edits?
3. There are no definitions provided for $f$ in Equations 5 and 6. Please clarify the meaning of these symbols.

**Questions For Authors:**

See weakness.

**Relation To Broader Scientific Literature:**

This paper contributes to the broader scientific literature by addressing the challenge of high-fidelity image manipulation through the technique of encoding target images into noise for editable manipulation.

**Theoretical Claims:**

No, as there are no theoretical claims made.

---

> ### Author Rebuttal · Authors · 2025-03-29
>
> We sincerely thank the reviewer for taking the time to evaluate our research. Below are our responses to all the points raised:
>
> **Paper Weaknesses:**
> > What does $Z_t^s$ represent in Figure 2? There is no clear definition of this symbol.
>
> **Answer:**
> We sincerely thank the reviewer for the detailed observations. In Figure 2, $Z_t^s$ represents the reconstructed latent. Methods such as Prompt-to-Prompt, Plug-and-Play, and MasaCtrl reconstruct the original image using this latent representation. We have updated Figure 2 to explicitly define each latent variable used in the illustration. Thank you.
>
> > The authors state that "smaller differences between the reconstructed and edited noise maps are strongly correlated with better editing performance." However, this is not always the case. As shown in Figure 3, the camel, elephant, and giraffe exhibit similar noise map differences, yet their editing performance varies. Have the authors attempted using the same editing prompt, such as "camel," to perform different types of edits?
>
> **Answer:**
> We acknowledge your concern regarding the relationship between noise map differences and editing performance. CLIPScore and LPIPS are based on deep learning models, which may introduce variations in perceived similarity even when noise map differences appear similar. We aimed to investigate the trend between noise map differences and editing performance. To do this, we conducted experiments on 20 different editing prompts, and as a result, the correlation coefficient between 'LPIPS / CLIPScore' and the 'difference between the reconstructed and edited noise maps' was found to be 0.8. Thank you for your careful review.
>
> > There are no definitions provided for $f$ in Equations 5 and 6. Please clarify the meaning of these symbols.
>
> **Answer:**
> The function $f$ is defined in Section 3.1 (Preliminaries, DDIM Inversion) of the paper as $z_{t-1} ← f(z_t, t, C)$. It is a function that calculates $z_{t-1}$ from $z_t$.
>
> We sincerely appreciate the reviewer for taking the time to review our research. We kindly ask you to consider our responses once more in your review. Thank you.

---

### Official Review · Reviewer_RLWR · 2025-03-14

**Overall Recommendation:** 3

**Summary:**

The paper introduces Editable Noise Map Inversion (ENM Inversion), a technique that improves both reconstruction quality and editing capabilities in diffusion-based image editing. ENM optimizes noise maps during inversion by minimizing the differences between reconstructed and edited versions, effectively encoding the target image's intended edits directly into the noise representation.

**Claims And Evidence:**

Yes.

**Essential References Not Discussed:**

Han, Ligong, Song Wen, Qi Chen, Zhixing Zhang, Kunpeng Song, Mengwei Ren, Ruijiang Gao et al. "Proxedit: Improving tuning-free real image editing with proximal guidance." In Proceedings of the IEEE/CVF Winter Conference on Applications of Computer Vision, pp. 4291-4301. 2024.

**Experimental Designs Or Analyses:**

The authors carefully selected appropriate baseline comparisons and established metrics to objectively measure editing performance and fidelity.

**Methods And Evaluation Criteria:**

The experimental design is sound, employing well-established benchmarks (PIE-Bench and DAVIS) alongside widely accepted evaluation metrics in diffusion-based image editing research (LPIPS, PSNR, SSIM, and CLIP similarity).

**Other Comments Or Suggestions:**

Additional visual examples showing challenging scenarios or failures of the proposed method could better illustrate practical limitations.

**Other Strengths And Weaknesses:**

The discussion of method efficiency is insufficient. While competing methods require only one inversion calculation per source image that can be applied across multiple target texts, the proposed method demands a separate inversion process for each target text and image combination—making it computationally expensive when performing multiple edits on a single image.

**Questions For Authors:**

Have you considered extending or comparing your approach to other generative modeling frameworks, such as flow-based models? If so, how does your method perform relative to such alternatives?

**Relation To Broader Scientific Literature:**

The key contributions build on existing diffusion inversion techniques, including DDIM, Null-Text Inversion, Negative Prompt Inversion, and Plug-and-Play methods.

**Theoretical Claims:**

This paper does not contain explicit theoretical proofs. It primarily contributes algorithmic and empirical insights rather than theoretical claims.

---

> ### Author Rebuttal · Authors · 2025-03-29
>
> First of all, we sincerely appreciate your time and effort in reviewing our research. Below, we provide responses to all the points raised.
>
> **Other Strengths And Weaknesses:**
> > The discussion of method efficiency is insufficient. While competing methods require only one inversion calculation per source image that can be applied across multiple target texts, the proposed method demands a separate inversion process for each target text and image combination—making it computationally expensive when performing multiple edits on a single image.
>
> **Answer:**
> Thank you for your insightful comments. It is indeed correct that our ENM Inversion method requires a separate inversion process for each text-image combination. This design choice was made to generate an optimal noise map tailored to each specific editing requirement. To compensate for efficiency while maintaining performance, we have carefully optimized our approach. As shown in Table 2, our method significantly reduces computational costs compared to NTI and StyleD in terms of inference time. This demonstrates that ENM Inversion can be a relatively efficient methodology for various image editing tasks. Once again, we sincerely appreciate your valuable feedback.
>
> **Essential References Not Discussed & Questions for Authors:**
> > Have you considered extending or comparing your approach to other generative modeling frameworks, such as flow-based models? If so, how does your method perform relative to such alternatives?
>
> **Answer:**
> Since our research focused on the inversion of diffusion models, we did not include a comparison with flow-based models. However, based on the reviewer’s suggestion, we conducted additional experiments incorporating ProxEdit and flow-based models [1][2], and we have included the results in the Appendix. Additionally, we integrated RF-Inversion into our method and conducted further experiments.
>
> | **Method** | **Structure Distance** ↓ | **PSNR** ↑ | **LPIPS** ↓ | **MSE** ↓ | **SSIM** ↑ | **CLIP Similarity (Whole)** ↑ | **CLIP Similarity (Edited)** ↑ |
> |--------------|----------------------------|------------|------------------|------------------|------------------|-------------------------|-------------------------|
> | ProxEdit   | 11.87                        | 27.12      | 45.70            | 31.70            | 85.73            | 24.13                   | 21.36                   |
> | SDEdit-Flux [1]  | 118.97                        | 14.41      | 329.92            | 450.06            | 60.82            | 25.06                   | 22.50                   |
> | RFInv [2]  | 60.08                        | 18.24      | 232.88            | 210.02            | 64.78            | 24.94                   | 22.65                   |
> | Ours + RFInv  | 46.38                        | 19.77      | 185.86            | 153.90            | 69.57            | 25.05                   | 22.65                   |
> | Ours  |  10.13                        | 28.19      | 45.26            | 27.02            | 86.29            | 25.30                   | 22.12                   |
>
> Table 1. Results of comparing and combining our method with flow-based models.
>
> The table above demonstrates that our approach outperforms editing methods using flow-based models in terms of structural distance, background preservation, and editability. Furthermore, integrating RF-Inversion into our method leads to even greater performance improvements.
>
> We thank the reviewer for engaging with us in the discussion.
>
> **Reference**
>
> [1] SDEdit: Guided Image Synthesis and Editing with Stochastic Differential Equations
>
> [2] Semantic Image Inversion and Editing using Rectified Stochastic Differential Equations

---

### Decision · Program_Chairs · 2025-05-01

**Decision:**

Accept (poster)

**Comment:**

This work has achieved unanimous positive reviews among reviewers in making sufficient contributions in diffusion model based image editing. The initial concerns are mostly around the evaluation and clarity (relatively minor), and the authors have addressed them during rebuttal.

Therefore I recommend acceptance but strongly encourage the authors to integrate the rebuttal into their revision.